

# Identification of quantitative trait loci (QTLs) regulating leaf SPAD value and trichome density in mungbean (*Vigna radiata* L.) using genotyping-by-sequencing (GBS) approach

Nikki Kumari[1,*], Gyan Prakash Mishra[1], Harsh Kumar Dikshit[1], Soma Gupta[1], Anirban Roy[2], Subodh Kumar Sinha[3], Dwijesh C. Mishra[4], Shouvik Das[1], Ranjeet R. Kumar[5], Ramakrishnan Madhavan Nair[6] and Muraleedhar Aski[1,*]

[1] Genetics, Indian Agricultural Research Institute, New Delhi, Delhi, India
[2] Plant Pathology, Indian Agricultural Research Institute, New Delhi, Delhi, India
[3] Biotechnology, National Institute of Plant Biotechnology, New Delhi, Delhi, India
[4] Agricultural Bioinformatics, Indian Agricultural Statistics Research Institute, New Delhi, Delhi, India
[5] Division of Biochemistry, Indian Agricultural Research Institute, New Delhi, Delhi, India
[6] ICRISAT, World Vegetable Center, South Asia, ICRISAT Campus, Patancheru, Hyderabad, India
[*] These authors contributed equally to this work.

Corresponding authors
Gyan Prakash Mishra,
gyan.gene@gmail.com
Harsh Kumar Dikshit, harshgenetic-siari@gmail.com

## ABSTRACT

Quantitative trait loci (QTL) mapping is used for the precise localization of genomic regions regulating various traits in plants. Two major QTLs regulating Soil Plant Analysis Development (SPAD) value (*qSPAD-7-1*) and trichome density (*qTric-7-2*) in mungbean were identified using recombinant inbred line (RIL) populations (PMR-1×Pusa Baisakhi) on chromosome 7. Functional analysis of QTL region identified 35 candidate genes for SPAD value (16 No) and trichome (19 No) traits. The candidate genes regulating trichome density on the dorsal leaf surface of the mungbean include *VRADI07G24840, VRADI07G17780*, and *VRADI07G15650,* which encodes for ZFP6, TFs bHLH DNA-binding superfamily protein, and MYB102, respectively. Also, candidate genes having vital roles in chlorophyll biosynthesis are *VRADIO7G29860, VRADIO7G29450*, and *VRADIO7G28520*, which encodes for s-adenosyl-L-methionine, FTSHI1 protein, and CRS2-associated factor, respectively. The findings unfolded the opportunity for the development of customized genotypes having high SPAD value and high trichome density having a possible role in yield and mungbean yellow vein mosaic India virus (MYMIV) resistance in mungbean.

## INTRODUCTION

Mungbean (*Vigna radiata* L.) is a leguminous crop, cultivated widely in various regions including Africa, South America, Australia, and Asian countries. It contains a diverse range of protein, fiber, antioxidants, and phytonutrients (*Reddy et al., 2021*). It is consumed in different forms like whole seeds, flour, sprouts, microgreens, *etc.*, making it an important source of dietary protein (*Priti et al., 2021*; *Priti et al., 2022*). The average World productivity of mungbean ranges between 2.5 and 3.0 t/ha, whereas mean Indian yield is only 0.5 t/ha (*All India Coordinated Research Project (AICRP), 2022*). This can be attributed to various factors, including biotic and abiotic constraints. Extensive research has been conducted on traits governing responses to various stresses, yet certain traits like soil plant analysis development (SPAD) value and trichome density have received limited attention in mungbean. On contrary, these have been extensively studied in soybean and other crops providing valuable insights into their genetic control and functional significance (*Yu et al., 2020*; *Angira et al., 2022*).

The leaf chlorophyll content is instrumental in light absorption and subsequent conversion of solar energy into chemical energy through photosynthesis (*Singhal et al., 2012*). It plays a crucial role in determining photosynthetic efficiency and yield (*Sakowska et al., 2018*). Traditionally, the assessment of leaf chlorophyll content involved extract-based quantification through spectrophotometric measurements of chlorophyll a and chlorophyll b (*Carter & Knapp, 2001*; *Dhanapal et al., 2016*). In contrast, SPAD method is a non-invasive, rapid, and cost-effective way of estimating the relative chlorophyll content using a portable SPAD chlorophyll meter (*Lombard et al., 2010*). A high SPAD value indicates a lower degree of photoinhibition during photosynthesis (*Yu et al., 2022*) and vice versa. A positive correlation between leaf chlorophyll content and seed yield was demonstrated in soybean, especially during reproductive stage (*Ma, Morrison & Voldeng, 1995*). However, such studies are lacking in mungbean.

Trichomes are specialized structures on the dorsal leaf surface that have significant intraspecific variability (*Chen et al., 2021*). Trichome density affects the insect feeding behavior, oviposition patterns, and nourishment of larvae (*Schillinger Jr & Gallun, 1968*; *Roberts et al., 1979*). Trichomes house specialized glands emitting terpenes, phenolics, alkaloids, and other olfactory and or gustatory deterrents which minimizes thrips and whiteflies infestation and thereby has a role in YMD resistance (*Kivimaki et al., 2007*; *Chen et al., 2021*; *Yasmin et al., 2022*).

Trichomes also acts as a physical barrier against biotic and abiotic stresses (*Kariyat et al., 2018*). In *Arabidopsis thaliana*, nearly 40 genes have been identified governing trichome density (*Mauricio, 2005*; *Atwell et al., 2010*). The key regulators include R2R3 MYB transcription factor GLABRA 1 (GL1), the basic helix-loop-helix (bHLH) transcription factor GLABRA 3/ENHANCER OF GLABRA 3 (GL3/EGL3), and the pleiotropic WD40 repeat protein TRANSPARENT TESTA GLABRA 1 (TTG1), collectively forming the MYB-bHLH-WDR (MBW) complex, a central component of the trichome initiation pathway (*Kirik et al., 2004*; *Zhao et al., 2008*). Significant progress has been made in understanding the genetic basis of trichome initiation and development in *Arabidopsis*, and soybean (*Li et*

*al., 2021*), however, research in mungbean remains very limited. Advancements in genetic mapping helped in the identification of robust single nucleotide polymorphism (SNP) markers (*Grant et al., 2011*) which helped in the construction of high-density genetic maps (*Qi et al., 2014*; *Jiang et al., 2020*).

This study aims to map the QTLs regulating SPAD value and trichome density in an RIL population derived from the cross between Pusa Baisakhi and PMR-1, which exhibit distinct responses for these traits. Pusa Baisakhi is characterized by low trichome density and light-colored leaves, while PMR-1 displays higher trichome density on the dorsal leaf surface and possesses darker leaves. The aim of this study was to map the QTLs regulating SPAD value and trichome density in mungbean using recombinant inbred line (RIL) population using GBS based genetic map (*Mathivathana et al., 2019*; *Chen et al., 2022*).

## MATERIAL AND METHODS

### Plant materials, mapping population, and trait measurement

The RIL population, consisting of 166 lines, was developed at the Indian Agricultural Research Institute (IARI) in New Delhi, India by crossing Pusa Baisakhi and PMR-1. Pusa Baisakhi is known for its low SPAD value (an indirect measurement of chlorophyll content) and low trichome density. On the other hand, PMR-1 exhibits a high SPAD value and high trichome density. To create the RIL population, the $F_1$ plants were self-pollinated to produce $F_2$ population and from $F_2$, 166 $F_{7:8}$ lines were developed using the single seed descent (SSD) method. This process involves selecting individual plants from each generation and allowing them to self-pollinate, followed by selecting a single pod from each plant for the next generation. Each RIL genotype was planted in a 4-meter row and spacing between each row and individual plants was maintained at 30 cm and 10 cm, respectively, allowing for approximately 40 plants in each row. From every row, five plants were randomly selected for the measurement of SPAD values and trichome density. The parents and RILs were screened for their SPAD values and trichome density during 2020 (July–November) and 2021 (July–November) in a randomized complete block design with two replications at the Indian Agricultural Research Institute (IARI), New Delhi, India (28.7041°N, 77.1025°E; 228.61 m above mean sea level). Field management followed the standard recommended mungbean cultivation practices for the area.

### SPAD value determination

The SPAD meter measures non-destructively the light transmittance of the leaf at 650 and 940 nm in the red and infrared wavelengths, providing a numerical output representing leaf greenness and chlorophyll concentration (*Markwell, Osterman & Mitchell, 1995*). The SPAD value was used as an indicator of the relative chlorophyll content in mungbean leaves. Five healthy plants situated in the middle of each row were randomly selected for SPAD value measurement using SPAD-502 Chlorophyll Meter at 45 DAS during 08:00 am to 11:00 am (*Wang et al., 2020*). The SPAD was measured at the top, middle, and bottom of a leaflet of the third leaf from the top in three replications (*Guang-Jun et al., 2010*). The mean SPAD value was designated as TSP (top site portion), MSP (middle site portion),

and BSP (bottom site portion), respectively. Additionally, the average SPAD values (ASP) were determined by calculating the mean of TSP, MSP, and BSP.

### Trichome density characterization

The trichome density was studied using five randomly selected plants per row by taking one fully expanded leaf (mid portion) at (i) seedling stage and (ii) after anthesis (*Chen et al., 2021*). The data was recorded using a compound microscope equipped with an ocular scale and bright field optics (10×). Before mounting on a glass slide, the upper epidermis (middle portion) of leaf was carefully coated with a thin layer of nail varnish and after five minutes, the film was peeled from the surface using a piece of transparent sticky tape. The scale of '0' (any short and sparse trichome) to '3' (long and dense trichome) was used for the measurement of trichome density.

### Statistical analysis

R software (*R Core Team, 2013*) was employed for comprehensive analysis. This encompassed ANOVA, Pearson phenotypic correlation computation, density plot assessment, and scatter plot visualization, Providing valuable insights into the data's variance, associations, distribution, and inter-variable relationships.

### DNA extraction and genotyping by sequencing

The genomic DNA isolated using cetyltrimethyl ammonium bromide (CTAB) method (*Lodhi et al., 1994*) from the young leaves of parental lines and 166 RIL population were used for the preparation of GBS libraries (*Elshire et al., 2011*). In brief, 100 ng DNA was digested for 4.0 h at 75 °C with ApeKI (New England Biolabs, Ipswitch, MA) in 20 µL volume containing 1 × NEB Buffer3 and 3.6U ApeKI. The purified 168-plex final DNA library was quantified using Bioanalyzer (Agilent Technologies) and was sequenced on Novaseq 6000 (Illumina® Inc., San Diego, CA, USA). Library construction and sequencing were done by NGB Diagnostics Pvt Ltd (India) and SNPs were identified. Variant calling was done using UGBS-GATK pipeline (version v3.6, https://gatk.broadinstitute.org/hc/en-us). Further SNPs were filtered based on minor allele frequency (MAF) of 5%, being present in at least 50% of the population, the proportion of heterozygosity, and polymorphism information,

### Genetic map construction and QTL analysis

The genetic linkage map of the RIL population was constructed based on the SNP markers identified through GBS and was used to map the QTLs for the SPAD value and trichome density traits. The high-quality SNPs were evaluated against the expected Mendelian segregation ratio using chi-square analysis for the genetic linkage map construction. The SNPs displaying distorted segregation ratios were excluded from the analysis and high-quality SNPs were used to construct the genetic map using JoinMap version 4.1 (*Stam, 1993*). A rippling algorithm having a window size of 1 was applied which improved the marker order by iteratively adjusting the marker positions within the linkage groups. To facilitate the comparative analysis, the marker distances were converted into centiMorgans (cM) using Kosambi's mapping function. Linkage groups (LGs) were visually represented

using Mapchart v2.32 (*Voorrips, 2022*). The order of SNPs on the genetic map was compared with their physical position on the mungbean reference genome downloaded from Ensembl Plants (release 54).

The QTL analysis was performed using composite interval mapping functions embedded in QTL IcIMapping v4.2 software (*Zeng, 1993*). To establish the statistical significance of the identified QTLs, a LOD threshold was calculated by performing 1000 permutations at $P \leq 0.05$ and phenotypic variance explained (PVE) value $\geq 10\%$. The nomenclature for QTL was like *qSPAD-7-1* where ''qSPAD'' represents the QTL for the SPAD value and 7-1 represents the first QTL on chromosome 7. For SPAD value, 142 RILs; while for trichome density, 166 RILs were used for the QTL analysis.

## Candidate gene and digital expression analysis

The genetic linkage map of the RIL (Pusa Baisakhi × PMR-1) was integrated with the physical map of the mungbean reference genome for the determination of physical locations of markers flanking the QTLs of interest. The mapping intervals of the detected QTLs were identified, and gene models falling within these intervals were retrieved from Ensembl Plants (https://plants.ensembl.org/biomart/martview), and the NCBI genome data viewer (GCF_000741045.1). In addition, other methods used include ShinyGO 0.77 analysis, DAVID, and gene annotation data. Genes known to be involved in SPAD value and trichome development, as reported in *Arabidopsis*, were collected from the *Arabidopsis* Information Resource (TAIR) (https://www.arabidopsis.org/index.jsp). The *Arabidopsis* genes were then used as queries in BLASTN search against the mungbean genome assembly to identify the candidate genes. Based on the well-characterized functions of these orthologous genes, candidate genes regulating SPAD value and trichome density were identified. Digital gene expression analysis was done using *Arabidopsis* orthologs and an Expression Angler was used to investigate their localized expression patterns (*Austin et al., 2016*; *Reddy et al., 2020*).

# RESULTS

## Evaluation of phenotypic variation

The parental genotypes and RILs displayed substantial variations in SPAD value and trichome density across two years (Table 1; Fig. S1) suggesting the presence of considerable diversity for these traits. Table 1 summarizes key descriptive statistics for SPAD value (chlorophyll content) and trichome density in parents (Pusa Baisakhi and PMR-1) and RIL populations. Some RILs showed lower or higher values than the parents for SPAD value and trichome density, indicating transgressive segregation during both the studied years. The difference between parents' SPAD values is 47.29, while trichome density ranges from 0 to 3. An analysis of the Pearson correlation coefficient indicated a non-significant positive correlation between SPAD value and trichome density ( $r^2 = 0.119$ ) in the $F_{7:8}$ RILs (Fig. 1).

## QTL identification for SPAD value and trichome

In total, 107.662 Gb raw data were generated and after cleaning 100.748 Gb high-quality data was utilized for downstream analysis. After adapter trimming, 98.33 Gb data was

**Table 1  Descriptive statistics for SPAD value (chlorophyll content) and trichomes in parents and RIL popuilation.**

| Trait | Parents | | Difference (parents) | RIL Population | | | |
|---|---|---|---|---|---|---|---|
| | | | | $F_{5:6}$ lines | | $F_{7:8}$ lines | |
| | Pusa Baisakhi | PMR-1 | | Range | Mean ± SD | Range | Mean ± SD |
| SPAD value | 2.39 | 49.68 | 47.29 | 1.75-43.66 | 24.01 ± 8.80 | 2.52-59.43 | 25.05 ± 9.9 |
| Trichome | 0 | 3 | 3 | 1-3 | 2.0 ± 0.59 | 1-3 | 2.11 ± 0.67 |

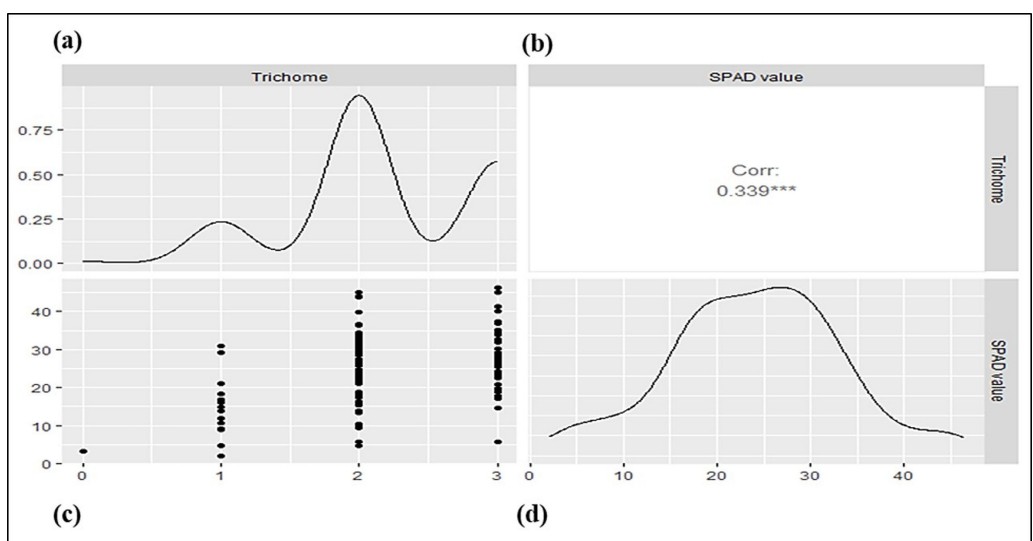

**Figure 1  The correlation between trichome density and SPAD values in a mungbean RIL population.**
(A) Density plot for SPAD value. (B) Pearson's phenotypic correlation between SPAD and trichome density. (C) Scatter plot. (D) Density plot for trichome density.

obtained from both parents and 166 RILs population which was used for final analysis. Initial screening yielded 6797 SNPs, and after applying various stringent selection criteria, the SNPs were reduced to 1,730 numbers. The genetic map was comprised of 1,730 SNPs, distributed across 11 linkage groups (LGs) ranging from 42.482 cM to 74.321 cM, and the marker density varied from 105 to 345. Overall, the genetic map covered a total genetic distance of 596.1 cM, with an average marker interval of 0.345 cM.

QTL analysis was conducted for SPAD value and trichomes using a linkage map having 1730 SNPs covering 11 chromosomes using inclusive composite interval mapping (ICIM) (Fig. 2). The results of the QTL analysis revealed one QTL for SPAD value and three QTLs for trichome density over two years (Table 2). The LOD scores of these QTLs ranged from 2.67 to 6.29 (Fig. 3), while the phenotypic variance explained ranged from 6.38 to 14.0%, indicating the proportion of phenotypic variation accounted for by the respective QTLs.

In the year 2020, ICIM analysis identified three QTLs related to SPAD value and trichome traits in an RIL population. A major QTL named, *qSPAD-7-1* (marker intervals VR7:55078511 and VR7:49782591) was found regulating the SPAD value on chromosome 7, which explained 11.55% of PVE. Two minor QTLs governing trichome density were

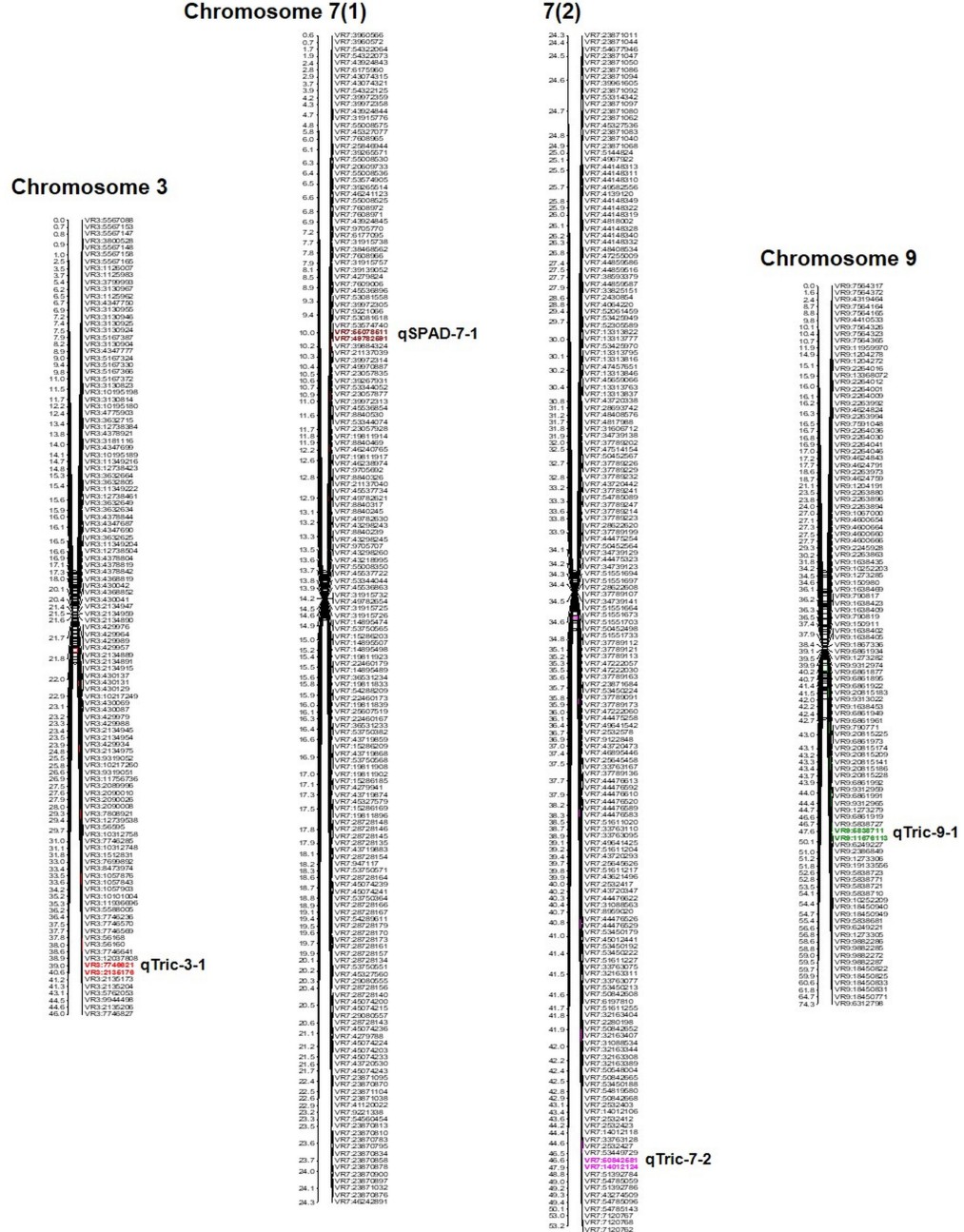

**Figure 2** Position of QTLs regulating trichome and chlorophyll content on chromosomes 03, 07, and 09 in a mungbean RIL mapping population.

identified on chromosome 3 (*qTric-3-1*; 6.38% PVE; marker interval VR3:7746621–VR3:2135176) and chromosome 9 (*qTric-9-1*; 6.74% PVE; marker intervals VR9:5838711 and VR9:11676113) (Fig. 2). The identified QTLs were found contributed by the parent PMR-1 for both SPAD value (*qSPAD-7-1*) and trichome density (*qTric-3-1*, *qTric-9-1*).

**Table 2  Details of QTLs governing SPAD value and trichomes in a mungbean RIL population.**

| Traits | QTL name | Year | LG[a] | Position[b] (cM) | Interval[c] (bp) | LOD[d] | PVE[e] (%) | A[f] |
|---|---|---|---|---|---|---|---|---|
| SPAD value | qSPAD-7-1 | 2020 | 7 | 10 | VR7:55078511-VR7:49782591 | 3.49 | 11.55 | 3.35 |
| | qSPAD-7-1 | 2021 | 7 | 10 | VR7:55078511-VR7:49782591 | 2.71 | 10.60 | 3.36 |
| | qSPAD-7-1 | Combined data | 7 | 10 | VR7:55078511-VR7:49782591 | 3.45 | 12.06 | 3.39 |
| Trichome | qTric-3-1 | 2020 | 3 | 39 | VR3:7746621-VR3:2135176 | 2.99 | 6.38 | 0.181 |
| | qTric-9-1 | 2020 | 9 | 48 | VR9:5838711-VR9:11676113 | 2.67 | 6.74 | 0.207 |
| | qTric-7-2 | 2021 | 7 | 47 | VR7:50842581-VR7:14012124 | 6.29 | 14.00 | −0.317 |
| | qTric-7-2 | Combined data | 7 | 47 | VR7:50842581-VR7:14012124 | 4.98 | 11.29 | −0.276 |

Notes.

LG[a], Linkage group; Position[b], Genetic position (cM) of QTL on linkage map; Interval[c], QTL LOD support interval; LOD[d], log of odds ratio at the peak likelihood of the QTL; PVE[e], Percentage of phenotypic variation explained by individual QTL; A[f], Additive effect.

On chromosome 7, a QTL for SPAD value (*qSPAD-7-1*; VR7:55078511 and VR7:49782591; 10.60% PVE) was identified (in 2020 and 2021), while a novel QTL for trichome density (*qTric-7-2*; VR50842581 to VR14012124; 14.0% PVE) was discovered in 2021 (Fig. 2).

The combined analysis revealed detection of two QTLs, namely *qSPAD-7-* 1 for SPAD value, and *qTric-7-2* for trichome density on chromosome 7 (Fig. 3). The *qSPAD-7-1* consistently appeared independently in both years (2020 and 2021) and also in the combined data with positive additive effects from PMR-1 for SPAD value. Whereas, *qTric-7-2* was detected in 2021 and in the combined data, with negative additive effects contributed by the parent Pusa Baisakhi.

## Gene ontology and candidate genes prediction of major QTLs

Two major QTLs (*qSPAD-7-1* and *qTric-7-2*) were chosen based on mapping results for gene ontology (GO) and candidate gene prediction analyses (Table S1). The genomic intervals of these QTLs were investigated, and 138 annotated genes were predicted within the 5.29 Mb physical interval of *qSPAD-7-1*, while 245 annotated genes were predicted within the 36.83 Mb interval of *qTric-7-2*. A set of 186 genes, of 383 genes located within the physical regions of two stable QTLs were selected based on the results of ShinyGO 0.77 analysis, gene annotation, and published literature and were used for further analysis (Table S2).

The study compared and validated 186 associated protein-coding genes using a BLASTN search against the *Arabidopsis* genome database. Of these, 35 genes exhibited high similarity (≥75%) to *Arabidopsis* genes and were extensively characterized (Table 3; Table S1). These genes regulate various diverse biological processes, including anatomical morphogenesis, porphyrin and chlorophyll metabolism pathways, and defense against viruses. Notably, most of the genes were from different families or coded by two genes, like Cytochrome b561 (*VRADI07G26610* and *VRADI07G09390*) and Protein trichome birefringence (*VRADI07G14170* and *VRADI07G23450*). Additionally, two genes (*VRADI07G30210* and *VRADI07G17680*) encode squamosa promoter-binding-like protein 13A, and two others (*VRADI07G10060* and *VRADI07G14740*) encode zinc finger proteins DOF5.8

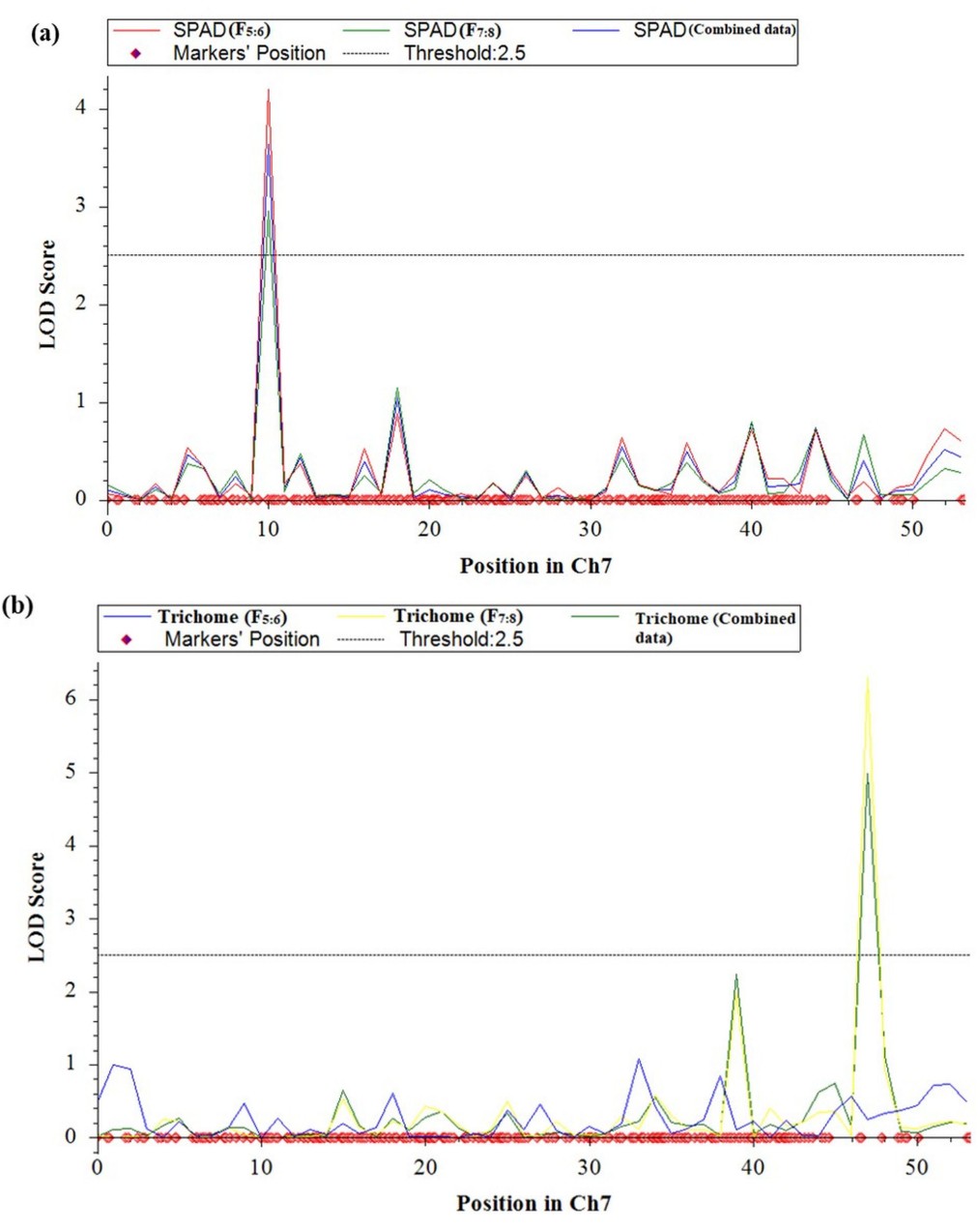

**Figure 3** LOD graphs of the identified QTLs for (A) SPAD value. (B) Trichome density derived from mungbean RILs (2020 and 2021) on the linkage maps.

and DOF5.6, respectively, belonging to the zinc finger DNA-binding domain family (Table 3).

Based on GO enrichment analysis, gene annotation, and published research on chlorophyll content, three genes were identified having a role in chlorophyll biogenesis (Table 3). These genes include *VRADI07G28520* which encodes CRS2 or Chloroplast

**Table 3  Details of candidate genes within two stable QTL regions for SPAD value and trichomes.**

| Name of QTL | Ensembl Gene ID | Position (Mbp) | Functional Annotation[a] |
|---|---|---|---|
| | VRADI07G26610 | 49.97 | Cytochrome b561 and DOMON domain-containing protein |
| | VRADI07G27270 | 50.58 | Glucose-1-phosphate adenylyltransferase large subunit 1, chloroplastic |
| | VRADI07G27460 | 50.83 | outer envelope pore protein 37, chloroplastic |
| | VRADI07G27620 | 51.10 | peptidyl-tRNA hydrolase, chloroplastic-like |
| | VRADI07G28180 | 51.65 | Probable glutathione peroxidase 2 |
| | VRADI07G28430 | 52.14 | NADPH:adrenodoxin oxidoreductase, mitochondrial |
| | VRADI07G28520 | 52.29 | CRS2-associated factor 1, chloroplastic-like |
| | VRADI07G28530 | 52.29 | Probable glycosyltransferase isoform X1 |
| | VRADI07G28660 | 52.40 | senescence/dehydration-associated protein , chloroplastic (LOC106767497) |
| qSPAD-7-1 | VRADI07G29130 | 52.86 | ABC transporter I family member 6, chloroplastic |
| | VRADI07G29450 | 53.17 | Probable inactive ATP-dependent zinc metalloprotease FTSHI 1, chloroplastic |
| | VRADI07G29740 | 53.44 | 6-phosphogluconate dehydrogenase, decarboxylating |
| | VRADI07G29860 | 53.57 | magnesium protoporphyrin IX methyltransferase, chloroplastic |
| | VRADI07G30110 | 53.92 | auxin-responsive protein SAUR71-like |
| | VRADI07G30210 | 54.07 | squamosa promoter-binding-like protein 13A |
| | VRADI07G31080 | 54.98 | Pentatricopeptide repeat-containing protein |
| | VRADI07G09390 | 24.96 | cytochrome b561 domain-containing protein |
| | VRADI07G10060 | 26.88 | Dof zinc finger protein DOF5.8 |
| | VRADI07G14170 | 33.81 | Protein trichome birefringence-like 12 |
| | VRADI07G14730 | 34.77 | DELLA protein RGL1-like (GRAS family) |
| | VRADI07G14740 | 34.79 | dof zinc finger protein DOF5.6 |
| | VRADI07G15650 | 36.20 | transcription factor MYB86-like |
| | VRADI07G16030 | 36.88 | NAC transcription factor 25-like |
| | VRADI07G16860 | 38.30 | F-box/kelch-repeat protein |
| | VRADI07G17610 | 38.98 | transcription factor TCP5 |
| | VRADI07G17620 | 38.98 | protein LONGIFOLIA 2-like isoform X1 |
| | VRADI07G17680 | 39.05 | Squamosa promoter-binding protein-like (SBP domain) TF family protein |
| | VRADI07G17740 | 39.12 | MPB2C |
| | VRADI07G17780 | 39.17 | basic helix-loop-helix (bHLH) DNA-binding superfamily protein |
| | VRADI07G19810 | 42.03 | zinc finger CCCH domain-containing protein 25 |
| | VRADI07G20260 | 42.56 | EPIDERMAL PATTERNING FACTOR-like protein 8 isoform X1 |
| | VRADI07G22030 | 44.89 | transcription factor RAX1 isoform X1 |
| | VRADI07G23450 | 46.56 | protein trichome birefringence-like 14 |

| Name of QTL | Ensembl Gene ID | Position (Mbp) | Functional Annotation[a] |
|---|---|---|---|
| | VRADI07G24840 | 48.19 | zinc finger protein 6-like |
| | VRADI07G24880 | 48.23 | COBRA-like protein 7 |

**Notes.**

[a]Candidate genes are identified using ShinyGO 0.77, Ensembl Plants, the legume information system (LIS), and available literature.

ribosomal protein S2-associated factor 1, chloroplastic-like protein belonging to YhbY-like superfamily imparting stability of chloroplast ndhA transcripts (*Li et al., 2021*); *VRADI07G29450* which encodes FTSHI1 or Filamentation Temperature-Sensitive H1 which links chloroplast biogenesis and division and belong to FtsH endopeptidase family clan MA(E) (*Kadirjan-Kalbach et al., 2012*); and *VRADI07G29860*, which encodes magnesium protoporphyrin IX methyltransferase and is involved in chlorophyll biosynthesis process and belong to S-adenosyl-L-methionine dependent methyl transferase superfamily.

For trichome development, a few transcription factors like basic Helix-Loop-Helix (bHLH), MYB102 (Myeloblastosis transcription factor 102), and Zinc finger proteins were identified. Of these, *VRADI07G24840* encodes Zinc Finger Protein 6 (ZFP6) which regulates trichome initiation through gibberellin and cytokinin signaling. The homologous gene models in *Arabidopsis*, including *AT4G25080, AT4G23940, AT5G08130*, and *AT4G21440* genes, were identified as orthologs of *VRADI07G29860, VRADI07G29450, VRADI07G17780*, and *VRADI07G15650*, respectively (Table S1).

The digital gene expression analysis revealed that the genes *AT4G25080, AT4G23940, AT5G08130, AT4G21440, AT5G47530, AT4G32360, AT2G43950, AT3G10670, AT4G18260, AT5G66940, AT1G18620* and *AT5G64470* which are the orthologs of *VRADIO7G29860, VRADIO7G29450, VRADIO7G17780, VRADIO7G15650, VRADI07G26610, VRADI07G28430, VRADI07G27460, VRADI07G29130, VRADI07G09390, VRADI07G10060, VRADI07G17620*, and *VRADI07G14170*, respectively; displayed highest expression in various plant tissues, including leaf, flower, shoot apex, young root, guard cell, trichomes, mesophyll cells, pollen tube, epidermis, etc. (Fig. 4). In addition, these genes also exhibited substantial expression in various organelles, including nucleus, mitochondria, endoplasmic reticulum, plastids, Golgi apparatus, and peroxisomes.

## DISCUSSION

QTL mapping is used to map the genomic regions regulating the studied traits using correlations between markers and the phenotypic traits in a mapping population (*Takuno, Terauchi & Innan, 2012*). A few studies have identified the loci governing chlorophyll content and trichome development in plants like Arabidopsis, tomato, and soybean (*Du, Yu & Fu, 2009*; *Fang et al., 2017*; *Wang et al., 2020*). However, studies on mungbean have remained scarce. This study is an attempt to find the QTLs associated with SPAD value (chlorophyll content) and trichome density traits in mungbean. An advanced RIL population ($F_6$ and $F_8$ generations) was used having 166 lines were used for the mapping

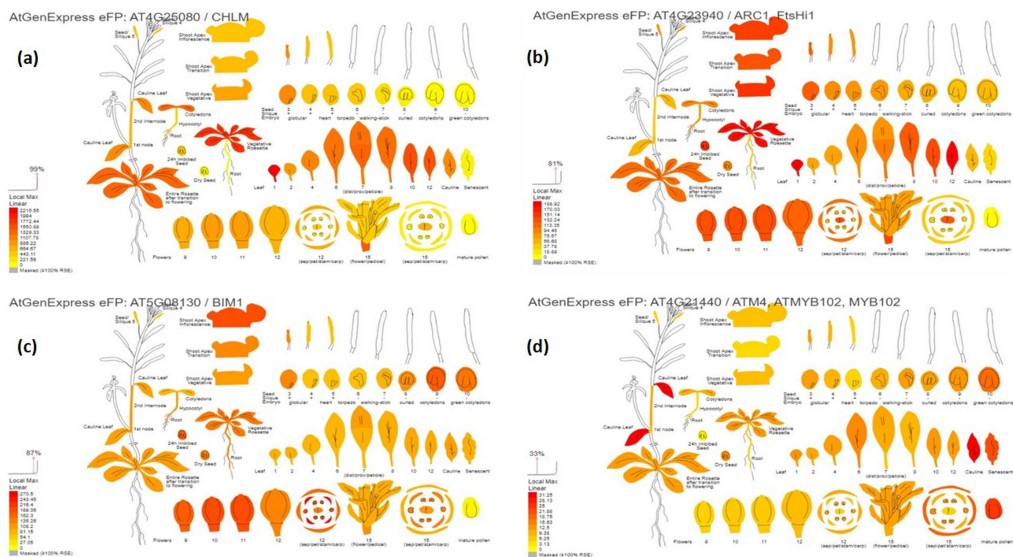

**Figure 4** **Digital gene expression patterns of identified candidate genes (depicted in Arabidopsis).** (A) CHLM (AT4G25080) encodes magnesium protoporphyrin IX methyltransferase (orthologous to VRA-DIO7G29860); (B) FtsHi1 (AT4G23940) is involved in chloroplast biogenesis and division (orthologous to VRADIO7G29450); (C) BIM1 (AT5G08130) encodes bHLH transcription factors and is involved in trichome development (orthologous to VRADIO7G29450); (D) ATM4 (AT4G21440) encodes MYB86 transcription factors regulating trichome branching and elongation (corresponds to VRADIO7G15650). Expression strength is color-coded: yellow for low, while red for high.

using GBS. These helped in the acquisition of precise and more reliable mapping of QTLs associated with the target traits.

## Variations in SPAD values and trichome traits

The RILs (Pusa Baisakhi × PMR-1) and their parents displaying significant differences in SPAD value and trichome density traits were used for precise mapping of the QTLs regulating these traits. No association was recorded between SPAD value and trichome density ($r^2 = 0.119$). Interestingly, the parents exhibited differential disease reactions (with Pusa Baisakhi being susceptible and PMR-1 resistant to MYMIV - *Mungbean Yellow Mosaic India Virus*). Trichomes do play a role in natural defense mechanisms against insects and in blackgram, a higher trichome density correlates with reduced whitefly presence (*Taggar & Gill, 2012*). Thus, there is a possibility that a RIL with high trichome density may confer enhanced whitefly tolerance and thereby MYMIV resistance to the mungbean. These findings highlight the necessity for further research to delve into the intricacies of chlorophyll content and trichome behavior in mungbean. SPAD is an indirect measure of chlorophyll content and higher chlorophyll may help, in the realization of more yield and also resistance to the MYMIV and vice versa. Previous studies have demonstrated the detrimental impact of MYMIV infection on SPAD values in mungbean (*Mishra et al., 2020*; *Dasgupta et al., 2021*).

## QTLs for chlorophyll and trichome traits and their implications

Chlorophyll and trichome traits are complicated quantitative traits that are also influenced by various environmental and hereditary factors (*Du, Yu & Fu, 2009*; *Wang et al., 2020*). A single significant QTL (*qSPAD-7-1*) for SPAD value in both generations ($F_{5:6}$ and $F_{7:8}$) and also in the combined data analysis suggests that this may be governed by a few major genes and also *via* quantitative inheritance (*Du, Yu & Fu, 2009*). It could be possible that the parental lines may carry the alleles contributing to the quantitative inheritance for chlorophyll content, but these may not be sufficiently diverse in the RILs to be detected as separate QTLs. In addition, the presence of epistatic interactions among multiple genes regulating SPAD value could have masked the individual effects of other QTLs, resulting in the detection of only one significant QTL (*Messmer et al., 2009*).

Four QTLs governing SPAD value and trichome traits have been identified. For trichome, *qTric-7-2* was mapped (RIL- $F_{7:8}$) and combined data showed a PVE range of 11.29% to 14.0%. Also, two more QTLs (*qTric-3-1* and *qTric-9-1*) have been detected ($F_{5:6}$) for trichomes. For SPAD value, only one significant, major, and stable QTL (*qSPAD-7-1*; PVE: 10.60% to 12.06%) was identified ($F_{5:6}$; $F_{7:8,}$ combined data). Several studies reported mapping of SPAD or chlorophyll-related QTLs in soybean on chromosome 7 (*Yu et al., 2020*; *Wang et al., 2020*). In *Barbarea vulgaris*, *Liu et al. (2019)* identified two QTLs (*qTric-4-1* and *qTri-8-1*) for trichomes on chromosomes 4 and 8, respectively; whereas, *Byrne et al. (2017)* reported it on chromosome 7. Leaf SPAD observations are collinearly correlated with leaf chlorophyll content for several crops (*Yadava, 1986*). It is important to note that in many studies, leaf chlorophyll values measured by SPAD chlorophyll meter were found to be positively associated with grain yield (*Maiti et al., 2004*; *Kandel, 2020*). All the identified QTLs of this study are novel and mapping of SPAD value and trichome density in mungbean will aid in identifying genetic markers, understanding resistance mechanisms, facilitating marker-assisted selection, and ultimately enhancing yield.

## Candidate gene analysis for SPAD value and trichomes

Identification of candidate genes is crucial for improving the target trait through the breeding approach. This study identified the candidate genes in the QTL region governing SPAD value (*qSPAD-7-1*) and trichomes (*qTric-7-2*) in mungbean. Out of the 186 genes extracted from the physical genomic interval of the two major QTLs, 35 were considered candidate genes. Network analysis has shown that the candidate genes for SPAD value are found associated with the biosynthesis of secondary metabolites, the pentose phosphate pathway, and porphyrin and chlorophyll metabolism (Fig. S2A). These genes contribute to chlorophyll formation by providing essential building blocks, coenzymes, antioxidants, and reducing power necessary for the synthesis and proper functioning of chlorophyll (*Pinto & Zempleni, 2016*; *Richter, Wang & Grimm, 2016*). Additionally, they may contribute to the plant's ability to defend against viral infections. A key candidate gene linked to *qSPAD-7-1* is *VRADI07G29860*, which is homologous to the *Arabidopsis AT4G25080* gene. This gene encodes a protein belonging to the S-adenosyl-L-methionine-dependent methyltransferase superfamily (*Richter, Wang & Grimm, 2016*). Magnesium-protoporphyrin IX methyltransferase converts magnesium-protoporphyrin IX

to magnesium-protoporphyrin IX metylester using S-adenosyl-L-methionine as a cofactor, which represents the second step in the biosynthesis of chlorophyll and bacteriochlorophyll from protoporphyrin IX.

Another important candidate gene is *VRADI07G29450*, which is homologous to the *Arabidopsis AT4G23940* gene. This encodes ATP-dependent zinc metalloprotease FTSHI1 protein, which serves as a target for several proteins involved in chlorophyll synthesis, including chlorophyllase and chlorophyll-a oxygenase (*Kadirjan-Kalbach et al., 2012*). By selectively degrading these proteins, FTSHI1 helps to maintain the balance between chlorophyll synthesis and degradation, ensuring the proper accumulation of chlorophyll in chloroplast. CRS2-associated factor 1, encoded by the *VRADI07G28520* gene (homolog of *Arabidopsis AT2G20020* gene) is a CAF1 RNA-binding CRS1/YhbY (CRM) domain-containing protein that is involved in the assembly and stabilization of chloroplast ribosomes. The chloroplast ribosome synthesizes proteins within the chloroplast, including those involved in chlorophyll biosynthesis (*Li et al., 2021*). Additionally, genes like *VRADI07G27270*, *VRADI07G27460*, *VRADI07G27620*, *VRADI07G28180*, and *VRADI07G28430*, share homology with *Arabidopsis* genes *AT5G19220*, *AT2G43950*, *AT1G18440*, *AT2G43350*, and *AT4G32360* which are involved in processes like glucose-1-phosphate adenylyltransferase activity, monoatomic ion channel activity, hydrolase activity, peroxidase activity, and oxidoreductase activity (*Goetze et al., 2006*; *Zybailov et al., 2008*; *Mugford et al., 2014*; *Attacha et al., 2017*; *Bellido et al., 2022*). These activities indirectly contribute to the overall cellular environment and metabolic processes necessary for chlorophyll synthesis (*Flores-Pérez & Jarvis, 2013*; *Cecchin et al., 2023*).

The candidate genes associated with trichomes are found associated with various processes like regulation of monopolar cell growth, cell morphogenesis, plant epidermis development, and defense response to viruses (Fig. S2B). The development and formation of trichomes are regulated by a complex network of genes and transcription factors. The *VRADI07G24840* gene (*AT1G10480* is *Arabidopsis* homolog) encodes a transcription factor ZFP5/ZFP6 which acts as the regulator of trichome initiation (*Zhou et al., 2013*). Molecular and genetic analyses suggest that ZFP6 functions upstream of ZFP8, ZFP5, and key trichome initiation regulators GL1 (GLABRA1) and GL3 (GLABRA3). ZFP6 and ZFP5 mediate the regulation of trichome initiation by integrating GA and cytokinin signaling in Arabidopsis.

Genes like *VRADI07G17780* and *VRADI07G15650*, which are homologous to the genes *AT5G08130* and *AT4G21440* in *Arabidopsis*, encode transcription factors such as basic helix-loop-helix (bHLH) family protein BIM1 (BES1-INTERACTING MYC-LIKE 1), involved in brassinosteroid signaling (*Liang et al., 2018*) and MYB102, involved in wounding and osmotic stress response (*Denekamp & Smeekens, 2003*). However, in *Arabidopsis*, other studies have shown that bHLH TFs, including GL3, ENHANCER OF GLABRA3 (EGL3), and TRANSPARENT TESTA GLABRA1 (TTG1), play a crucial role in trichome formation (*Bernhardt et al., 2003*). These factors form a protein complex known as the TTG1-GL3/EGL3-GL1 (TTG) complex, which regulates the transcriptional control of trichome development. The TTG complex functions as a positive regulator of trichome

formation by activating the expression of downstream genes involved in trichome initiation and growth.

MYB102, a MYB TF, has also been reported in *Arabidopsis*, interact with bHLH proteins to promote trichome development (*Hao et al., 2021*). MYB102 is specifically expressed in developing trichomes and acts downstream of the bHLH complex. It functions as a positive regulator of trichome branching and elongation. Other genes like *VRADI07G20260*, *VRADI07G24880*, *VRADI07G23450*, *VRADI07G14730*, and *VRADI07G17620*, which are homologous to *Arabidopsis* gene models *AT1G61120*, *AT3G16860*, *AT5G20680*, *AT2G01570*, and *AT1G18620*, contribute to the complex regulatory network underlying trichome development. They influence various aspects such as patterning, elongation, differentiation, and cell wall organization (*Attaran, Rostás & Zeier, 2008*; *Parsons et al., 2012*; *Wang et al., 2014*; *Lee et al., 2018*).

Furthermore, these candidate genes were validated using digital gene expression analysis. For SPAD value, *AT4G25080*, *AT4G23940*, and *AT2G20020*, displayed the highest expression in leaves, flowers, mesophyll, shoot apex, and various parts of the inflorescence (Figs. 4A and 4B). For trichome density, *AT1G10480*, *AT5G08130*, and *AT4G21440*, showed the highest expression in leaves, flowers, young roots, guard cells, trichomes, pollen tubes, and epidermis (Figs. 4C and 4D). *Reddy et al. (2020)* also validated a few key genes in mungbean through digital gene expression analysis for phosphorus use efficiency traits. In the future, by targeted manipulation of identified candidate genes *via* genetic engineering or gene editing tools, desired changes in chlorophyll content and trichome density can be induced for the incorporation of MYMIV resistance and enhanced yield.

## CONCLUSIONS

This study comprehends the genetic foundations of traits like SPAD value (chlorophyll content) and trichome density, in mungbean. The QTL mapping has identified the genomic regions governing SPAD value and trichome density traits. The study identified 35 candidate genes in two major QTLs viz., *qSPAD-7-1* and *qTric-7-2* which govern SPAD value and trichome density in mungbean, respectively. The identified QTLs can be used for trait enhancement through dedicated breeding efforts. The mapped QTL for trichome density spanned a distance of nearly 36 Mb which prompts consideration for future investigations, where efforts may be directed towards refining the resolution of this mapping to pinpoint the specific gene governing this trait.

## ACKNOWLEDGEMENTS

The technical support received by Mr. Dilip Kumar (Indian Agricultural Research Institute, New Delhi) is duly acknowledged.

### Funding

This research was funded by the Indian Council of Agricultural Research (ICAR), New Delhi, and SERB (Science and Engineering Research Board), New Delhi (CRG/2019/002024). Ramakrishnan M. Nair received funding from the long-term strategic donors of the World Vegetable Center: Taiwan, the United States Agency for International Development (USAID), the UK Government's Foreign, Commonwealth & Development Office (FCDO), the Australian Centre for International Agricultural Research (ACIAR), Germany, Thailand, Philippines, Korea, Japan, and funding from the ACIAR Project on International Mungbean Improvement Network (CROP/2019/144). The funders had no role in study design, data collection and analysis, decision to publish, or preparation of the manuscript.

### Grant Disclosures

The following grant information was disclosed by the authors:
Indian Council of Agricultural Research (ICAR), New Delhi.
SERB (Science and Engineering Research Board), New Delhi: CRG/2019/002024.
World Vegetable Center: Taiwan.
The United States Agency for International Development (USAID).
The UK Government's Foreign, Commonwealth & Development Office (FCDO).
Australian Centre for International Agricultural Research (ACIAR).
ACIAR Project on International Mungbean Improvement Network: CROP/2019/144.

### Competing Interests

The authors declare there are no competing interests.

### Author Contributions

- Nikki Kumari performed the experiments, analyzed the data, prepared figures and/or tables, authored or reviewed drafts of the article, and approved the final draft.
- Gyan Prakash Mishra conceived and designed the experiments, authored or reviewed drafts of the article, and approved the final draft.
- Harsh Kumar Dikshit conceived and designed the experiments, authored or reviewed drafts of the article, and approved the final draft.
- Soma Gupta performed the experiments, prepared figures and/or tables, and approved the final draft.
- Anirban Roy performed the experiments, analyzed the data, prepared figures and/or tables, and approved the final draft.
- Subodh Kumar Sinha analyzed the data, prepared figures and/or tables, and approved the final draft.
- Dwijesh C. Mishra analyzed the data, prepared figures and/or tables, and approved the final draft.
- Shouvik Das performed the experiments, analyzed the data, prepared figures and/or tables, and approved the final draft.

- Ranjeet R. Kumar analyzed the data, prepared figures and/or tables, and approved the final draft.
- Ramakrishnan Madhavan Nair conceived and designed the experiments, authored or reviewed drafts of the article, and approved the final draft.
- Muraleedhar Aski conceived and designed the experiments, analyzed the data, authored or reviewed drafts of the article, and approved the final draft.

## DNA Deposition

The following information was supplied regarding the deposition of DNA sequences:

The sequence data are available at GenBank: PRJNA914895.

## Data Availability

The raw data for two-year SPAD values and Trichome data are available in the Supplemental Files.

## Supplemental Information

Supplemental information for this article can be found online at http://dx.doi.org/10.7717/peerj.16722#supplemental-information.

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
