# Peer review of "Identification of quantitative trait loci (QTLs) regulating leaf SPAD value and trichome density in mungbean (Vigna radiata L.) using genotyping-by-sequencing (GBS) approach"

_PeerJ, doi:10.7717/peerj.16722_

## Round 0.1 · original submission · Major Revisions

The authors are requested to revise the manuscript as per the reviewers' comments.

**Language Note:** The review process has identified that the English language must be improved. PeerJ can provide language editing services - please contact us at copyediting@peerj.com for pricing (be sure to provide your manuscript number and title). Alternatively, you should make your own arrangements to improve the language quality and provide details in your response letter. – PeerJ Staff

Reviewer 1 ·

Basic reporting

The manuscript “ Identiûcation of quantitative trait loci (QTLs) governing SPADbased chlorophyll content and leaf trichomes in mungbean (Vigna radiata L.) using genotyping by sequencing (GBS) approach” deals with the The study investigates the mapping of quantitative trait loci (QTLs) associated with SPAD value and trichome density in mungbean, with the aim of better understanding these traits and their relevance to yellow mosaic disease resistance. While the research addresses an important aspect of mungbean breeding, several points need to be addressed to improve the quality and clarity of the manuscript.
Comments
• The manuscript could benefit from a clearer structure and organization, particularly in the Materials and Methods, Results, and Discussion sections. Ensure that the flow of information is logical and easy for readers to follow.
• Define any abbreviations and technical terms upon first use to improve readability
• It is important to explicitly state the main research objectives or questions at the beginning of the manuscript. What specific questions or hypotheses is the study trying to address regarding SPAD value and trichome density?
• Provide more detailed information on the experimental setup, including the size of the RIL populations, the number of replicates, and the statistical methods used for QTL mapping.
• Describe the criteria used for selecting the candidate genes for functional analysis. Were there any bioinformatic tools or databases employed?
• Mention the specific software or algorithms used for QTL mapping and candidate gene identification.
• Clearly present the results of QTL mapping, including the chromosomal locations, confidence intervals, and effect sizes of the identified QTLs.
• Include graphical representations (e.g., genetic maps, QTL plots) to illustrate the genomic locations of the identified QTLs.
• For the functional analysis, provide details on the annotation of candidate genes, such as their putative functions, biological pathways, or known roles in other plant species.
• Quantify the phenotypic variation explained (PVE) by each QTL and discuss their significance.
• Interpret the results in the context of mungbean breeding and disease resistance. Why are SPAD value and trichome density important traits for yellow mosaic disease resistance?
• Discuss the potential practical applications of the identified QTLs and candidate genes for mungbean breeding programs.
• Address limitations and potential sources of variation in the study, as well as avenues for further research.
• Summarize the key findings and their implications concisely in the conclusion section.
• Ensure that all references are accurately cited and listed following a standardized citation style.
• Carefully proofread the manuscript to eliminate language and grammar issues, which can affect the overall readability.

Experimental design

Please see the comments in section 1

Validity of the findings

Please see the comments in section 1

Additional comments

Please see the comments in section 1

Reviewer 2 ·

Basic reporting

The study has the potential to contribute significantly to our understanding of overall crop growth and yellow mosaic disease resistance in mungbean. There are certain suggestions for further improvement of this manuscript.
1. Please modify the title as “governing leaf trichomes” is not giving clear indication of studied parameters.
2. Write the practical application instead of “development of customized genotypes”.
3. In abstract authors have mentioned “key traits regulating yellow mosaic disease in mungbean” while in introduction “Understanding the genetic control of SPAD value and trichome density in mungbean is crucial for targeted breeding efforts to improve photosynthetic efficiency, plant resilience, and overall crop productivity. This raises a question about what was the main objective for which this study was planned?
4. The study should provide adequate background about yellow mosaic disease in mungbean and why the traits under study are key in regulating it. This will provide context to readers unfamiliar with the subject.

Experimental design

5. The study measures SPAD values at the top, middle, and bottom of plants. Different positions on a plant might have varying chlorophyll concentrations due to light exposure and age of the leaves. While authors have taken an average, this may not fully capture the variability. Please comment.

Validity of the findings

6. There are several typos in the text. Please proofread the document thoroughly to correct these errors.
7. Expand the discussion on the significance of the identified candidate genes. Discuss the implications of these genes in the context of mungbean development and resistance to yellow mosaic disease.
8. Consider referencing prior studies or evidence that support the roles of these candidate genes.
9. Ensure that all relevant studies and sources are adequately cited, particularly those that might have informed the methodology or provided context for the results.

---

## Round 0.2 · accepted · Accept

The manuscript has been revised as per the suggested comments and now it can be accepted in its current form.

Reviewer 1 ·

Basic reporting

The authors made substantial changes in the manuscript.

Experimental design

The authors made substantial changes in the manuscript.

Validity of the findings

The authors made substantial changes in the manuscript.

Additional comments

The authors made substantial changes in the manuscript.

Reviewer 2 ·

Basic reporting

The manuscript was improved as per the suggestions.

Experimental design

Design is appropriate

Validity of the findings

All the findings are valid.